# Modulation of Gut Microbial Community and Metabolism by *Bacillus licheniformis* HD173 Promotes the Growth of Nursery Piglets Model

**DOI:** 10.3390/nu16101497

**Published:** 2024-05-15

**Authors:** Jiaxuan Li, Cheng Tian, Shuaifei Feng, Wei Cheng, Shiyu Tao, Changchun Li, Yuncai Xiao, Hong Wei

**Affiliations:** 1Key Laboratory of Agricultural Animal Genetics, Breeding and Reproduction of the Ministry of Education, Huazhong Agricultural University, Wuhan 430070, China; lijiaxuan@webmail.hzau.edu.cn (J.L.); tiancheng0301@126.com (C.T.); fengshuaifei@webmail.hzau.edu.cn (S.F.); 2Key Laboratory of Swine Genetics and Breeding of the Ministry of Agriculture, Huazhong Agricultural University, Wuhan 430070, China; 3College of Animal Sciences and Technology, Huazhong Agricultural University, Wuhan 430070, China; chengwei87924@163.com (W.C.); sytao@mail.hzau.edu.cn (S.T.); 4National Key Laboratory of Agricultural Microbiology, Huazhong Agricultural University, Wuhan 430070, China; 5College of Veterinary Medicine, Huazhong Agricultural University, Wuhan 430070, China

**Keywords:** *Bacillus licheniformis* HD173, nursery piglets, gut microbiota, butyric acid, gut metabolites

## Abstract

Maintaining the balance and stability of the gut microbiota is crucial for the gut health and growth development of humans and animals. *Bacillus licheniformis* (*B. licheniformis*) has been reported to be beneficial to the gut health of humans and animals, whereas the probiotic effects of a new strain, *B. licheniformis* HD173, remain uncertain. In this study, nursery piglets were utilized as animal models to investigate the extensive impact of *B. licheniformis* HD173 on gut microbiota, metabolites, and host health. The major findings were that this probiotic enhanced the growth performance and improved the health status of the nursery piglets. Specifically, it reduced the level of pro-inflammatory cytokines IL-1β and TNF-α in the serum while increasing the level of IL-10 and SOD. In the gut, *B. licheniformis* HD173 reduced the abundance of pathogenic bacteria such as *Mycoplasma*, *Vibrio*, and *Vibrio metschnikovii*, while it increased the abundance of butyrate-producing bacteria, including *Oscillospira*, *Coprococcus*, and *Roseburia faecis*, leading to an enhanced production of butyric acid. Furthermore, *B. licheniformis* HD173 effectively improved the gut metabolic status, enabling the gut microbiota to provide the host with stronger metabolic abilities for nutrients. In summary, these findings provide scientific evidence for the utilization of *B. licheniformis* HD173 in the development and production of probiotic products for maintaining gut health in humans and animals.

## 1. Introduction

Trillions of microbes reside in the gut lumen of animals, forming an intricate ecosystem known as the gut microbiota, which performs a pivotal role in diet nutrient metabolism and sustaining host health via regulating gut physiological functions through their own actions and metabolites [1,2,3]. Extensive investigations into germ-free animals devoid of gut microbiota have unveiled significant physiological and immunological disparities when compared to their normal counterparts. Notably, these disparities encompass compromised development of the gastrointestinal tract epithelium, diminished organ and lymph node sizes, and profound alterations in the spectrum of immune cell types and the serum’s proteomic and metabolomic landscapes, all attributed to the absence of gut microbiota. These pivotal studies reinforce the indispensable role that gut microbiota plays in the integral physiological development of animal organisms [4]. The gut microbiota within animals is perpetually in a state of dynamic flux, and maintaining a stable, balanced ecosystem is pivotal for sustaining normal physiological and biochemical operations in the intestine. This intricate network interacts symbiotically with the host, playing a fundamental role in fostering intestinal development, reinforcing the intestinal barrier, optimizing metabolic processes and nutritional equilibrium, and modulating host immunity [5,6,7]. Conversely, imbalance or disruption within gut microbiota can have deleterious effects, leading to intestinal damage, dysfunction, and ultimately, the manifestation of inflammation and disease [2,8].

Given the paramount importance of gut microbiota in the lifecycle of humans and animals, it is imperative to maintain the balance of this microbial community through nutritional interventions, as it holds significant benefits for the overall health and well-being of both humans and animals. Nursery piglets, having undergone the transition from breastfeeding to weaning and experiencing changes in environmental factors, are prone to disturbances in their gut microbiota, which allows pathogenic and opportunistic bacteria to invade and cause intestinal dysfunction. Hence, they serve as an excellent model for exploring microecological disorders and intestinal dysfunction [9,10,11]. In recent years, an increasing number of probiotics have been discovered, demonstrating advantages in promoting gut development, maintaining immune balance, and regulating the gut microecosystem [12,13,14].

*Bacillus licheniformis* (*B. licheniformis*) is a bacterium that is widely used in various fields such as biomedicine, food industry, animal husbandry, and agriculture, and it possesses significant research value and is considered an effective probiotic for maintaining animal gut health [15,16]. Studies have indicated that certain strains of *B. licheniformis*, such as *B. licheniformis* S6, can modulate the intestinal microbiota, enhance the intestinal barrier, and promote intestinal immunity [15,17,18,19,20]. Furthermore, *B. licheniformis*, a spore-forming bacterium, possesses the remarkable ability to persist through spores in adverse environmental conditions [21]. Extensive studies have demonstrated that this bacterium can successfully navigate the harsh conditions of the animal stomach, germinate, and proliferate in the small intestine to exert its beneficial effects [22]. Owing to these distinctive traits, *B. licheniformis* holds immense potential for the development of commercially viable probiotic products [23]. The safety of using *B. licheniformis* as a probiotic has been verified in various studies, and products made from *B. licheniformis* have obtained safety approval in Europe, the United States, Japan, and other regions [24,25,26].

However, previous reports have described varying mechanisms of *B. licheniformis* in its interaction with the animal gut, and different strains within the same species may have different biological effects [27]. Therefore, the impact and mechanisms of different *B. licheniformis* subtypes on the gut remain worthy of further investigation. A newly identified strain, *B. licheniformis* HD173, has yet to be studied, and we primarily aim to reveal the probiotic effects of this strain by comparing nursery piglets fed with a diet supplemented with *B. licheniformis* HD173 to those consuming a regular diet in terms of growth performance, serum inflammatory cytokines, oxidative stress indicators, and gut microbiota and metabolites. We aspire for this study to furnish scientific evidence that will underpin the utilization of *B. licheniformis* HD173 in the development of probiotic products that significantly contribute to the enhancement of gut health in humans and animals.

## 2. Materials and Methods

### 2.1. Experimental Animals and Dietary Treatments

A total of 54 Duroc × Landrace × Yorkshire crossbred piglets (27 barrows and 27 gilts, weaned at 21 ± 3 d of age) at an age of 35 ± 3 d (average initial body weight 8.8 ± 0.08 kg) were divided into 3 treatments and provided ad libitum access to feed and water. The 3 treatments were as follows: control group (a basal diet), 0.02% BL group (a basal diet containing 200 g/t of *B. licheniformis* HD173 powder), and 0.04% BL group (a basal diet containing 400 g/t of *B. licheniformis* HD173 powder). Feed was given for 28 days, and the body weights in each group were weighed on 0 d, 14 d, and 28 d. The average daily weight gain (ADG) was calculated. The test strain *B. licheniformis* HD173 used in this study was screened and preserved in Hua Da Rui Er Co., Ltd. (Wuhan, China). The dry powder of *B. licheniformis* HD173 contained live bacteria at a concentration of 1.2 × 10^11^ CFU/g. Diets for nursery piglets were formulated based on the National Research Council (2012) and detailed in Appendix A. Antibiotic drugs were not used during the experiment. The animal experimental and sample collection procedures were approved by the Institutional Animal Care and Use Committee of Huazhong Agricultural University (Wuhan, China). All animal protocols used in this study were in accordance with the Health Guidelines for the Care and Use of Laboratory Animals at Huazhong Agricultural University.

### 2.2. Sample Collections

On 28 d, after overnight fasting, 6 nursery piglets from each group, whose weights were close to the average weight of the group, were selected for sample collection. The nursery piglets were immobilized for blood collection from the jugular vein. After allowing the blood to settle, it was centrifuged to obtain serum. The anus areas of the selected nursery piglets were thoroughly cleaned and disinfected. Subsequently, sterile cotton swabs were used to stimulate their anus in order to promote defecation. Immediately after excretion, the fresh feces were collected in sterile cryopreservation tubes.

### 2.3. Enzyme-Linked Immunosorbent Assay

Commercial kits (Bioswamp, Wuhan, China) were used to analyze inflammatory cytokines and oxidative stress indicators in the serum of nursery piglets. Inflammatory cytokines include tumor necrosis factors-α (TNF-α, Cat # PR40133), interleukin-1β (IL-1β, Cat # PR40098), interleukin-6 (IL-6, Cat # PR40121), and interleukin-10 (IL-10, Cat # PR40137). Oxidative stress indicators include superoxide dismutase (SOD, Cat # PR40023) and malondialdehyde (MDA, Cat # PR40147). The specific test procedures follow the manufacturer’s protocol.

### 2.4. DNA Extraction, 16S rRNA Gene Amplicon and Sequencing

Total genomic DNA samples were extracted using the OMEGA Soil DNA Kit (M5636-02) (Omega Bio-Tek, Norcross, GA, USA) following the manufacturer’s instructions and stored at −20 °C prior to further analysis. The quantity and quality of extracted DNAs were measured using a NanoDrop NC2000 spectrophotometer (Thermo Fisher Scientific, Waltham, MA, USA) and agarose gel electrophoresis, respectively. PCR amplification of the bacterial 16S rRNA genes V3–V4 region was performed using the forward primer 338F (5′-ACTCCTACGGGAGGCAGCA-3′) and the reverse primer 806R (5′-GGACTACHVGGGTWTCTAAT-3′). PCR amplicons were purified with Vazyme VAHTSTM DNA Clean Beads (Vazyme, Nanjing, China) and quantified using the Quant-iT PicoGreen dsDNA Assay Kit (Invitrogen, Carlsbad, CA, USA). After the individual quantification step, amplicons were pooled in equal amounts, and pair-end 2 × 250 bp sequencing was performed using the Illlumina NovaSeq platform with NovaSeq 6000 SP Reagent Kit (500 cycles) at Shanghai Personal Biotechnology Co., Ltd. (Shanghai, China).

Microbiome bioinformatics were performed with QIIME2 2019.4 [28] with slight modification according to the official tutorials. Briefly, raw sequence data were demultiplexed using the demux plugin followed by primers cutting with the cutadapt plugin [29]. Sequences were then quality-filtered, denoised, and merged and had their chimera removed using the DADA2 plugin [30]. Non-singleton amplicon sequence variants (ASVs) were aligned with mafft [31]. Alpha-diversity metrics (Chao1 [32], Observed species, Shannon [33], Simpson [34]) and beta-diversity metrics (Bray distance) were estimated using the diversity plugin. Alpha diversity was shown by R package ggplot2 (v3.4.1). Beta diversity was analyzed using principal co-ordinate analysis (PCoA) based on Bray distance. Permutational multivariate analysis of variance (PERMANOVA) was applied by R package vegan (v2.6.4) to explore the significance of the differentiation of microbiota structure among groups with 999 permutations. Taxonomy was assigned to ASVs using the classify-sklearn naïve Bayes taxonomy classifier in the feature-classifier plugin [35] against the Greengenes Database. Histograms of taxonomy composition at the phylum and genus level were drawn with the R package ggplot2. Gut microbial markers were determined by linear discriminant analysis effect size (LEfSe) using the online Huttenhower Galaxy server. Random forest analysis was established by qiime2 and shown by the R packages pheatmap and ggplot2. Spearman correlation between differential gut microorganisms was analyzed with the R package corrplot.

### 2.5. Quantitative Analysis of Short-Chain Fatty Acids (SCFAs)

SCFAs of feces (acetic acid, propionic acid, isobutyric acid, butyric acid, isovaleric acid, valeric acid, and hexanoic acid) were estimated using a gas chromatographic (GC) approach. Briefly, the samples were thawed on ice, and 30 mg of each sample was taken and placed in a 2 mL centrifuge tube. Then, 50 μL of 20% phosphoric acid was added to resuspend the sample. Subsequently, 4-methylpentanoic acid was added to a final concentration of 500 μg/mL as an internal standard, and the mixture was shaken vigorously for 2 min. After centrifugation at 14,000× *g* for 20 min, the supernatant was collected and transferred to a sample vial for GC-MS analysis. The injection volume was 1 μL, and the split ratio was set to 10:1 for split injection. The chromatographic peak areas and retention times were extracted using the MSD ChemStation software (vB.02.01). Standard curve graphs were plotted, and the content of short-chain fatty acids in the samples was calculated.

### 2.6. Untargeted Metabolomics Analysis

Fecal samples were slowly thawed at 4 °C, and 1 mL precooling methylalcohol/acetonitrile/water (2:2:1, *v*/*v*) was added and adequately vortexed. After ultrasonic decomposition at low temperature for 30 min, the samples are incubated for 10 min at −20 °C to precipitate the protein and then centrifuged (14,000× *g*, 4 °C, 20 min). The supernatants were collected and dried under vacuum and then stored at −80 °C standby. We redissolved the sample in 100 μL acetonitrile/water (1:1, *v*/*v*) and adequately vortexed it before centrifuging (14,000× *g*, 4 °C, 15 min). The supernatants were collected for LC-MS/MS analysis.

Sample separation was performed using an UHPLC (1290 Infinity LC, Agilent Technologies, USA) HILIC and RPLC. QC samples were inserted into the analysis queue to evaluate the system stability and data reliability during the whole experimental process. The mobile phase of chromatography was composed of buffer A (water +25 mM ammonium acetate +25 mM ammonium hydroxide) and buffer B (acetonitrile). Samples were detected in both ESI positive and negative modes. Analyses were performed using an UHPLC coupled to a quadrupole time-of-flight (TripleTOF 6600, AB SCIEX, USA). The ESI source conditions following HILIC separation were set. The raw MS data (wiff.scan files) were converted to MzXML files using ProteoWizard (v3.0.8789) MS Convert and processed using XCMS (v3.12.0) for feature detection, retention time correction, and alignment. The metabolites were identified by accuracy mass (<25 ppm) and MS/MS data which were matched with our standards database. In the extracted ion features, only the variables having more than 50% of the nonzero measurement values in at least one group were kept.

Positive and negative datasets were imported into R software (v4.3.1). After normalization, principal component analysis (PCA) and orthogonal partial least-squares-discriminant analysis (OPLS-DA) were carried out to visualize metabolic alterations between two groups using the R package ropls (v1.22.0). Variable importance in projection (VIP) ranks the overall contribution of each variable to the OPLS-DA model. Variables with VIP > 1.0, fold change > 2, and *p* < 0.05 Student’s *t*-test scores were considered differential metabolites. The volcano plots were used to show differential metabolites using the R package ggplot2 (v3.4.1). The abundance table of differential metabolites was used to construct a random forest classification model using mlr3verse software (v0.2.7). Spearman correlations between marker microorganisms and fecal metabolites, inflammatory cytokines, and oxidative stress indicators were analyzed with the R package psych.

### 2.7. Statistical Analysis

The experimental data were expressed in the form of ‘Mean ± SEM (standard error of the mean)’. The differences between the three groups was analyzed by ANOVA (analysis of variance). Multiple comparisons were performed to compare the significance of differences between groups using Tukey’s correction. Differences between two groups were determined with the Student’s *t*-test or Mann–Whitney U Test using R software (V4.3.1). The threshold of significance was set at *p* < 0.05.

## 3. Results

### 3.1. The Effects of B. licheniformis HD173 on the Growth Performance of Nursery Piglets

In order to evaluate whether the *B. licheniformis* HD173 can influence the growth performance of young animals, we checked the growth performance of nursery piglets, the data are shown in Appendix A. According to the growth performance results in Figure 1A, the body weight of the 0.02% BL group showed a significant increase at 28 d compared to the control group. From 0 d to 14 d, the ADG of both 0.02% BL group and 0.04% BL group showed a significant increase compared to the control group (Figure 1B). During the period from 15 d to 28 d, the ADG of only 0.02% BL group was higher than control group (Figure 1B). In the overall process from 0 d to 28 d, there was a significant difference in ADG between the 0.02% BL group and the control group (Figure 1B). However, there was no significant difference in ADG between 0.04% BL group and the control group (Figure 1B).

### 3.2. The Effects of B. licheniformis HD173 on Level of Serum Inflammatory Cytokines and Oxidative Stress Indicators

To evaluate the effect of *B. licheniformis* HD173 on inflammation and oxidative stress in nursery piglets, we measured the content of TNF-α, IL-1β, IL-6, IL-10, SOD, and MDA in serum using an ELISA kit. Both the 0.02% BL and 0.04% BL groups had significantly lower levels of IL-1β than the control group (Figure 2A). Meanwhile, the concentration of TNF-α showed a significant decrease compared to the control group (Figure 2B). The level of IL-6 was not significantly different between the three groups (Figure 2C). At the level of IL-10, both the 0.02% BL and 0.04% BL groups were significantly higher than the control group (Figure 2D). SOD and MDA respond to a certain extent to the level of oxidative stress in animals. The result showed that the levels of SOD in 0.02% BL and 0.04% BL groups were significantly increased compared to that in the control group (Figure 2F). Furthermore, the level of MDA in the 0.02% BL group showed a reduced trend compared to the control group (Figure 2E).

### 3.3. The Effect of B. licheniformis HD173 on the Gut Microbiota of Nursery Piglets

According to the results above, 0.02% *B. licheniformis* HD173 supplementation appears to be more effective for the growth and health of nursery piglets. Hence, the fecal samples from 0.02% BL group and the control group were used for further investigations. To explore the effect of *B. licheniformis* HD173 on gut microbiota, six samples from the 0.02% BL group and six samples from the control group were subjected to 16S rRNA amplicon sequencing. The Observed_species, Chao1, Shannon, and Simpson indexes of fecal microbiota in 0.02% BL group were not significantly different from those in the control group (Figure 3A). However, according to the PCoA based on the bray distances, the samples of 0.02% BL group were distinguished from the control group in microbial community structure, which suggested that the microbial composition of 0.02% BL group and the control group were different (Figure 3B). The relative abundances of the top-10 most abundant phyla and the top-20 most abundant genera were shown in Figure 3C,D. At the phylum level, the Firmicutes and Bacteroidetes were the dominant phylum in both two groups. At the genus level, the microbiota of 0.02% BL group was mainly composed of *Prevotella*, *Oscillospira*, *Treponema*, and *Lactobacillus*, whereas *Prevotella*, *Treponema*, *Oscillospira*, and *Paludibacter* were the dominant genera of the control group.

To determine the marker microorganisms, the statistical differences of microbial abundance in feces were evaluated by using the LEfSe, A total of 39 marker microorganisms were identified from taxonomic phylum to species. In the 0.02% BL group, eight genera and five species were enriched, while in the control group, five genera and two species were enriched (Figure 3E). Random forest is a classical and efficient machine learning algorithm based on decision trees; it is able to classify microbial community samples effectively, robustly, and accurately. A random forest classification model was constructed with relative abundance at the genus level and species level from the two groups of nursery piglets, and the 20 genera and 20 species with the highest level of importance are shown in Appendix A. These communities at the top of the importance scale can be regarded as marker microorganisms between the BL group and the control group. Those communities that were identified as marker microorganisms by both two analytical methods were analyzed in detail. Compared with the control group, the 0.02% BL group showed significantly increased abundances of *Oscillospira*, *Coprococcus*, and *Roseburia faecis* (Figure 3F); these communities were reported in previous studies to be associated with the production of SCFAs, especially butyric acid. At the same time, *Mycoplasma*, *Vibrio*, and *Vibrio metschnikovii* were reduced in relative abundance (Figure 3F) and are potential pathogens in the gut.

### 3.4. The Effects of B. licheniformis HD173 on the Production of SCFAs

As mentioned above, we found that the 0.02% BL group was enriched with some communities related to the production of SCFAs; therefore, we measured the content of SCFAs in fecal samples to evaluate the effects of *B. licheniformis* HD173 on SCFAs. In the determination of SCFAs, the butyrate contents of 0.02% BL group were significantly higher than the control group, and the contents of acetic acid and total SCFAs in 0.02% BL group showed an increased trend (Figure 4A). We analyzed the correlation of short-chain fatty acid content with inflammatory cytokines, oxidative stress indicators, and identified marker microorganisms. The results showed that increased butyric acid content was most positively correlated with *Oscillospira*, *Corynebacterium*, *Coprococcus*, and *Corynebacterium stationis* and was most negatively correlated with *Vibrio*, *Vibrio metschnikovii*, and *Ruminococcus callidus* (Figure 4B). In addition, the content of IL-10 in serum was positively correlated with butyric acid (Figure 4B).

### 3.5. The Effects of B. licheniformis HD173 on the Gut Metabolites of Nursery Piglets

The gut microbiota plays a crucial role in the digestion and absorption of nutrients, thereby influencing gut metabolism. To examine the impact of *B. licheniformis* HD173 on the gut metabolites of nursery piglets, an LC-MS/MS-based untargeted metabolomic analysis was conducted to determine the changes in fecal metabolomic profiles with *B. licheniformis* HD173 intervention. The PCA analysis confirmed the system stability and data reliability during the entire experimental process, as all QC samples were clustered together (Figure 5A,B). Meanwhile, the samples in the 0.02% BL group showed a clear distinguishment from the samples in the control group, which indicated that the metabolic profiles of the two groups were different (Figure 5A,B). Furthermore, the OPLS-DA models in positive and negative ionization modes were built, and the results showed that the two groups of samples could be clearly distinguished (Figure 5C,D). Subsequently, the validity and robustness of the OPLS-DA model were assessed by a permutation test; the intercept between the regression line of Q2 and the vertical axis was less than 0, which indicated that neither model was overfitted and both were able to explain the real differences in gut metabolites between the two groups (Appendix A).

Differential metabolites between the two groups were determined using the VIP values of the OPLS-DA model, the *p*-values of the Student’s *t*-test, and fold change values. A total of 148 metabolites were up-regulated in the BL group, while 31 metabolites were down-regulated in the 0.02% BL group compared to the control group (Figure 5E). According to the abundance of the differential metabolites in the two groups, we constructed a random forest classification model and identified the 20 metabolites with the highest level of importance (Figure 5F). Notably, among the metabolites enriched in the 0.02% BL group we found some beneficial substances such as albiflorin and pamidronic acid.

KEGG pathway enrichment analysis of differential metabolites revealed differences in gut microbial function between the two groups of nursery piglets. Differential metabolites are mainly enriched in the following pathways: amino acid metabolism (glycine, serine, and threonine metabolism; phenylalanine metabolism; alanine, aspartate, and glutamate metabolism; and phenylalanine, tyrosine, and tryptophan biosynthesis), carbohydrate metabolism (galactose metabolism, ascorbate and aldarate metabolism, and pentose phosphate pathway), xenobiotic degradation and metabolism (atrazine degradation, nitrotoluene degradation, and styrene degradation), lipid metabolism (secondary bile acid biosynthesis and arachidonic acid metabolism) (Figure 5G).

### 3.6. Correlations between Gut Microbiota and Inflammatory Cytokines, Oxidative Stress Indicators, Gut Metabolites

We then explored the correlation between the gut microbiota with gut metabolites, inflammatory cytokines, and oxidative stress indicators. Figure 6 showed a positive correlation between the relative abundance of 9 bacteria upregulated in the 0.02% BL group and the relative abundance of 19 metabolites also upregulated in the 0.02% BL group, as well as a positive correlation with the levels of IL-10 and SOD. On the other hand, the relative abundance of six bacteria downregulated in the 0.02% BL group show a positive correlation with the levels of IL-1β, TNF-α, and SOD but a negative correlation with the level of IL-10. These results indicate that there is a close relationship between gut microbiota and metabolites, as well as the physiological indicators of the host.

## 4. Discussion

In recent years, many studies have shown that probiotics can promote human and animal growth and improve animal health, and *B. licheniformis* is one of the widely used probiotics. Previous studies have highlighted the prominent role of *B. licheniformis* in reducing diarrhea rates, enhancing immunity and resistance, and improving growth performance [36,37,38,39]. In this study, none of the nursery piglets, including those in the control group, exhibited notable diarrhea symptoms, therefore, *B. licheniformis* HD173 did not demonstrate a pronounced effect in reducing the diarrhea rate. Consistent with previous research findings, we observed that the supplementation of *B. licheniformis* HD173 into the diet reduced the level of the pro-inflammatory cytokines IL-1β and TNF-α in the serum of nursery piglets, while simultaneously increasing the level of anti-inflammatory cytokine IL-10. Meanwhile, the supplementation of *B. licheniformis* HD173 also increased the level of SOD in the serum, and there was a tendency for the MDA content to decrease. These discoveries indicate that the *B. licheniformis* HD173 strain has the potential to enhance the anti-inflammatory and antioxidant stress capabilities of nursery piglets. In terms of growth performance, the supplementation of 0.02% *B. licheniformis* HD173 significantly promoted the ADG of nursery piglets. Although piglets receiving 0.04% *B. licheniformis* HD173 exhibited a higher ADG during the first half of the experiment, it did not show a significant increase compared to the control group in the latter half. This indicates that a lower dose of *B. licheniformis* HD173 is more effective in promoting weight gain in nursery piglets, which is consistent with the research results of Yu et al. [18]. These results suggests that the effect of *B. licheniformis* HD173 is not entirely dose-dependent, and excessive accumulation of *B. licheniformis* HD173 strains does not necessarily yield better results; however, the underlying mechanisms remain to be further explored. In addition, due to limitations in experimental conditions (such as the availability of automatic feed recording equipment), we were unable to obtain accurate measurements of feed intake and feed conversion ratios, and this aspect requires further improvement and refinement in future studies.

One of the well-known functions of probiotics is their ability to modulate the gut microbiota of animals. By facilitating the colonization of beneficial bacteria in the gut or suppressing the growth of harmful bacteria, probiotics contribute to the maintenance of a balanced state of the gut microbiota, which is crucial for improving the gut health of the host [40,41,42,43]. Recent studies have demonstrated the regulatory effect of *B. licheniformis* HD173 on gut microbiota. Wang et al. [37] demonstrated in their study on weaned piglets that the *B. licheniformis* mix increased Simpson’s diversity index in the gut microbiota. However, our data showed that *B. licheniformis* HD173 did not increase the α-diversity indexes such as Shannon, Simpson, and Chao1. Our results demonstrate that the gut microbial composition of nursery piglets underwent significant changes after dietary supplementation with *B. licheniformis* HD173, which is consistent with several other studies on *B. licheniformis* [18,36,43]. The abundance of *Mycoplasma*, *Vibrio*, and *Vibrio metschnikovii* decreased significantly in the *B. licheniformis* HD173 supplementation group, and they have been reported as potential pathogens in some studies [44,45]. Moreover, we found that the abundance of these pathogenic was significantly positively correlated with the level of pro-inflammatory cytokines IL-1β and TNF-α. We speculate that the lower level of pro-inflammatory cytokines in the nursery piglets of the *B. licheniformis* HD173 supplementation group are associated with the reduced abundance of these potential pathogens. On the other hand, our results showed that the abundance of some beneficial bacteria, such as *Oscillospira*, *Coprococcus*, and *Roseburia faecis*, increased after supplementation with *B. licheniformis* HD173. *Oscillospira* has consistently been detected in the gut or feces of humans and animals through 16S rRNA amplicon sequencing and metagenomic sequencing, it occupies a significant proportion of the gut microbiota in animals and is believed to be closely related to host health [46,47]. Several studies have shown a negative correlation between the abundance of *Oscillospira* with obesity, inflammation, and metabolic diseases [48,49,50]. Researchers have reconstructed nearly complete genomes of *Oscillospira* from metagenomic data. By analyzing its genomic features through sequence similarity, gene neighborhood information, and manual metabolic pathway curation, it has been inferred that *Oscillospira* is a butyrate producer [51]. In addition, *Coprococcus* and *Roseburia faecis* have also been identified as important butyrate producers in the animal gut [52,53]. The increase in the abundance of these butyrate-producing bacteria suggests that the production of butyrate may have increased, which we confirmed with subsequent SCFA measurements showing higher butyrate levels in the feces after supplementation with *B. licheniformis* HD173. Moreover, our results indeed demonstrated a significant positive correlation between butyrate level and the abundance of butyrate-producing bacteria, as well as a significant positive correlation between IL-10 levels and butyrate. Butyric acid is an organic acid produced by specific bacteria in the gut lumen through fermentation of mainly undigested dietary carbohydrates, and it is the main energy source for colon cells and plays a beneficial role in cellular energy metabolism and gut balance [54,55]. Butyric acid primarily serves as a histone deacetylase inhibitor and binding to several specific G protein-coupled receptors to regulate the physiological responses of the host gut [56,57]. Its positive roles in anti-inflammation, modulation of gut immunity, and maintenance of the gut mucosal barrier have been revealed [58,59,60]. In conclusion, *B. licheniformis* HD173 increased the abundance of butyrate-producing bacteria in the guts of nursery piglets, thereby promoting the production of butyric acid, and this may be one of the mechanisms underlying its ability to enhance the anti-inflammatory ability of hosts.

Changes in the composition of gut microbiota can lead to alterations in the metabolites they produce, ultimately resulting in changes in the functional roles of the gut microbiota. In this study, non-targeted metabolomics analysis revealed that, compared to the control group, nursery piglets receiving *B. licheniformis* HD173 supplementation exhibited significantly increased concentrations of 148 metabolites and significantly decreased concentrations of 31 metabolites. These findings suggest that the changes in gut microbiota induced by *B. licheniformis* HD173 also led to extensive alterations in gut metabolism. Consistent with the increase in the abundance of beneficial bacteria in the gut, we observed an elevation in the level of several beneficial metabolites, such as albiflorin and pamidronic acid, in the *B. licheniformis* HD173 supplementation group. In some studies, albiflorin has been reported to possess the ability to alleviate inflammatory damage, exhibiting prominent effects in anti-inflammatory and immune regulation [61,62,63]. Pamidronic acid is a substance that exhibits beneficial effects on bone development and is effective in the treatment of osteolytic bone metastases and osteoporosis [64,65]. The increased level of these beneficial metabolites can enhance the anti-inflammatory ability of nursery piglets and promote their growth and development. KEGG pathway enrichment analysis revealed that most of the differential metabolites were enriched in the metabolic pathways of amino acids, carbohydrates, and lipids, which are fundamental nutrients. These findings indicate that the mediation of *B. licheniformis* HD173 may enable the gut microbiota to possess stronger nutritional metabolic abilities, which in turn effectively assists the host in extracting energy from food and promoting the host’s growth and development.

## 5. Conclusions

In summary, the supplementation of *B. licheniformis* HD173, a novel subtype of *B. licheniformis*, improved the growth performance of nursery piglets and enhanced the immune status of the host by reducing the production of pro-inflammatory cytokines and oxidative substances and increasing the production of anti-inflammatory cytokines and antioxidants. Moreover, we have also demonstrated that *B. licheniformis* HD173 can effectively modulate the gut microbiota and their metabolites. This strain reduced the abundance of potential pathogenic bacteria in the gut of nursery piglets while it increased the abundance of beneficial bacteria, thereby promoting the production of beneficial metabolites such as butyric acid and improving intestinal metabolism. These findings provide scientific evidence for the utilization of *B. licheniformis* HD173 in the development and production of probiotic products for maintaining gut health in humans and animals.

## Figures and Tables

**Figure 1 nutrients-16-01497-f001:**
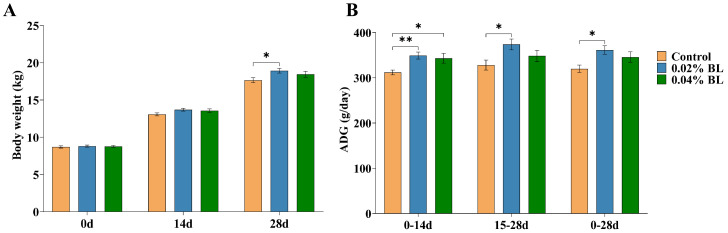
*B. licheniformis* HD173 effect on growth performance in nursery piglets. (**A**) Body weight of nursery piglets at 0 d, 14 d, and 28 d; (**B**) ADG of nursery piglets at 0–14 d, 15–28 d, and 0–28 d. *n* = 18. * indicates the degree of significant difference (*p* < 0.05); ** indicates the degree of significant difference (*p* < 0.01).

**Figure 2 nutrients-16-01497-f002:**
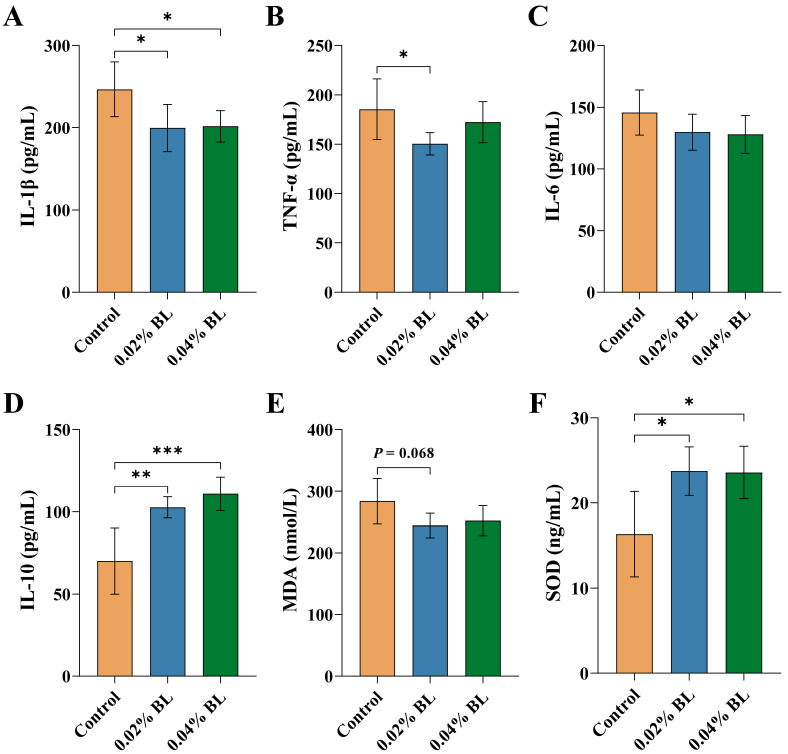
*B. licheniformis* HD173 effect on level of serum inflammatory cytokines and oxidative stress indicators. (**A**) IL-1β level, (**B**) TNF-α level, (**C**) IL-6 level, (**D**) IL-10 level, (**E**) MDA level, and (**F**) SOD level. *n* = 6. * indicates the degree of significant difference (*p* < 0.05); ** indicates the degree of significant difference (*p* < 0.01); *** indicates the degree of significant difference (*p* < 0.001).

**Figure 3 nutrients-16-01497-f003:**
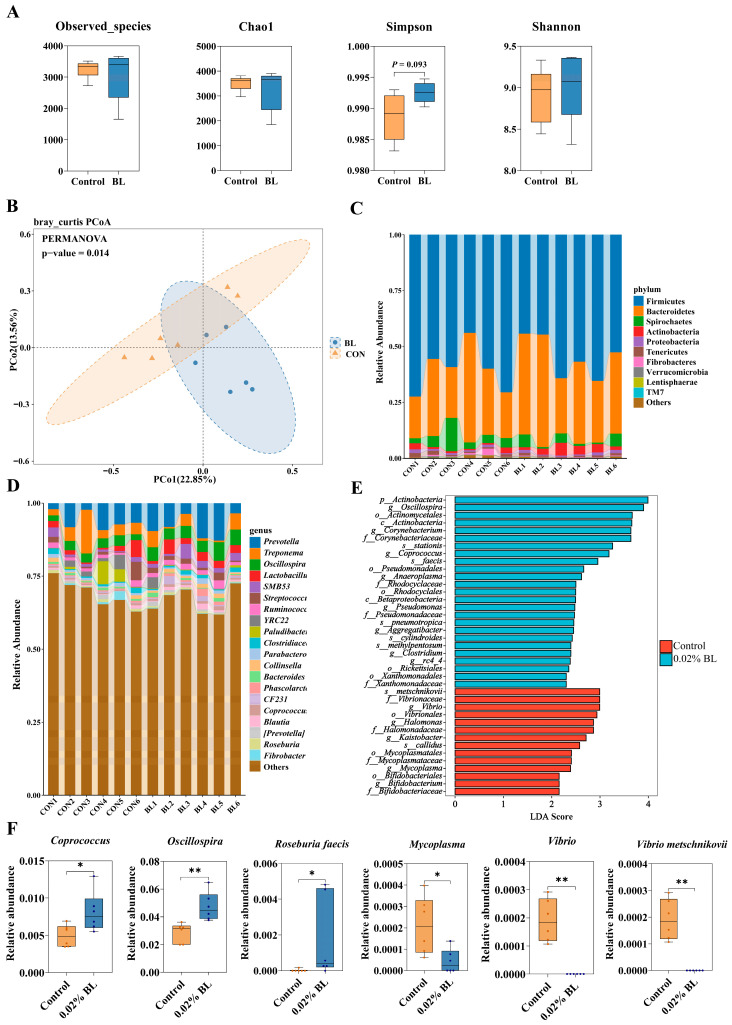
*B. licheniformis* HD173 effect on the gut microbiota of nursery piglets. (**A**) The Observed_species, Chao1, Shannon, and Simpson indexes; (**B**) PCoA based on Bray–Curtis distances; (**C**) the relative abundance of gut microbiota at phylum level; (**D**) the relative abundance of gut microbiota at genus level; (**E**) discriminant analysis of multi-level species differences by LEfSe analysis from phylum to species level; and (**F**) relative abundance of *Coprococcus*, *Oscillospira*, *Roseburia faecis*, *Mycoplasma*, *Vibrio*, and *Vibrio metschnikovii*. *n* = 6. * indicates the degree of significant difference (*p* < 0.05); ** indicates the degree of significant difference (*p* < 0.01).

**Figure 4 nutrients-16-01497-f004:**
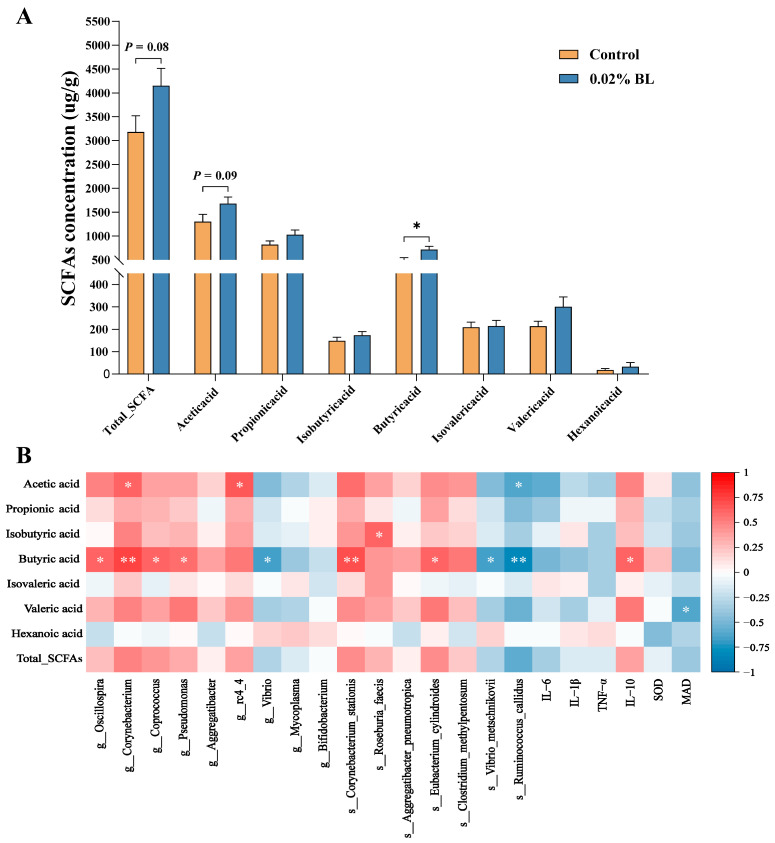
The effects of *B. licheniformis* HD173 on the production of SCFAs. (**A**) The level of acetic acid, propionic acid, isobutyric acid, butyric acid, isovaleric acid, valeric acid, hexanoic acid, and total SCFAs; (**B**) correlations between level of SCFAs and gut microbiota, inflammatory cytokines, and oxidative stress indicators. *n* = 6. * indicates the degree of significant difference (*p* < 0.05); ** indicates the degree of significant difference (*p* < 0.01).

**Figure 5 nutrients-16-01497-f005:**
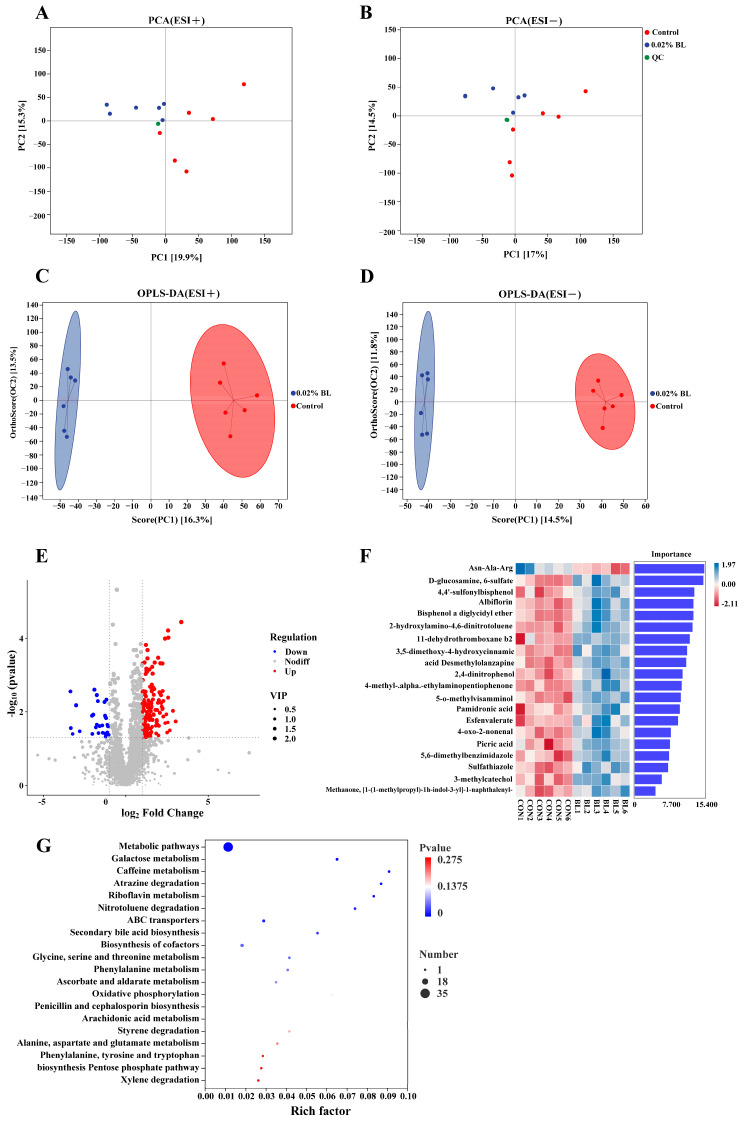
Effects of *B. licheniformis* HD173 on metabolome composition of feces in nursey piglets. (**A**,**B**) The score plot of PCA in positive (ESI+) and negative ionization modes (ESI−), (**C**,**D**) the score plot of OPLS-DA in positive (ESI+) and negative ionization modes (ESI−), (**E**) the volcano plot analysis of the differential metabolites, (**F**) the random forest model analysis of the differential metabolites, and (**G**) bubble plot showing the KEGG enrichment analysis. *n* = 6.

**Figure 6 nutrients-16-01497-f006:**
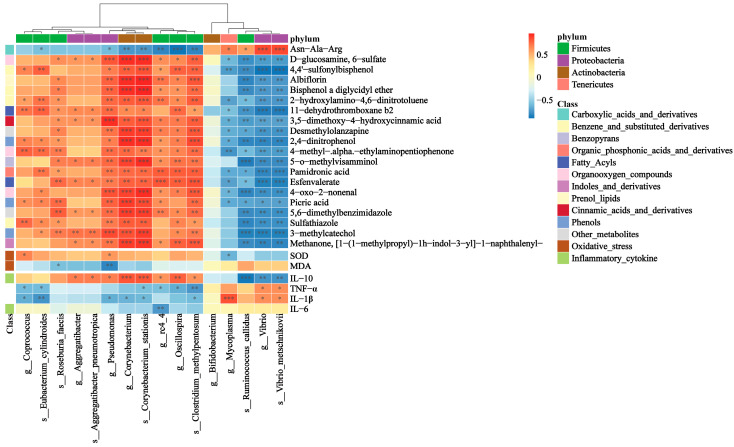
Heatmap of the Spearman’s rank correlation coefficient between the gut microbiota and gut metabolites, inflammatory cytokines, and oxidative stress indicators. *n* = 6. * indicates the degree of significant difference (*p* < 0.05); ** indicates the degree of significant difference (*p* < 0.01); *** indicates the degree of significant difference (*p* < 0.001).

## Data Availability

The data presented in this study are available on request from the first author or corresponding author.

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
