# Peer review of "Modulation of Gut Microbial Community and Metabolism by Bacillus licheniformis HD173 Promotes the Growth of Nursery Piglets Model"

_nutrients, 2024, doi:10.3390/nu16101497_

Round 1

Reviewer 1 Report

Comments and Suggestions for Authors

The article entitled Modulation of Gut Microbial Community and Metabolism by Dietary Bacillus licheniformis HD173 Supplementation May promote the growth of Nursery Piglets Model by Jiaxuan Li in which they aimed to analyze the nursery piglets as animal model to investigate the extensive  impact of B. licheniformis HD173, a new strain, on gut microbiota, metabolites and host health.

The manuscript is based on a good idea but I am surprised that how the editor sent it for review with 43% of Similairty index. The authors must work on this manuscript and reduce the plagarisim upto 17-18%.

I would recommend a major revision of this manuscript.The authors must revise the manuscript according to the following comments.

Abstract:

Add conclusion to the abstract.

Introduction:

Line 36-38 Trillions of microbes inhabit in the gut lumen and constitute a complex ecosystem, 36 the gut microbiota, which plays a crucial role in diet nutrient metabolism and maintaining host health by regulating the physiological functions of the gut through themselves and metabolites [1-3].

References are of 2020, 2012 and 2010.

The authors should add recent literature and references as there are thousands of articles published in the past 5-6 years which provide insightful information. The following articles would definitely improve the introduction and discussion section of the manuscript. The authors must read and cite them.

1.     Lacticaseibacillus paracasei BNCC345679 Revolutionises DSS Induced-Colitis and Modulates Gut Microbiota. 10.3389/fmicb.2024.1343891

2.     Genome investigation and Functional Annotation of Lactiplantibacillus plantarum YW11 Revealing Streptin and Ruminococcin-A as Potent Nutritive Bacteriocins against Gut Symbiotic Pathogens.

Line 38-44 can also be improved by read the above mentioned articles.

Line 57-58. Mention the specific strain

The methodology is good.

Results are well explained

Figures’ quality is good.

Discussion is nicely written

Conclusion should be revised

Reviewer 2 Report

Comments and Suggestions for Authors

The manuscript titled "Modulation of Gut Microbial Community and Metabolism by..." by Jiaxuan Li is quite interesting and shows promise. However, it requires significant revisions before publication. I believe it has potential but needs improvement.

Below are my comments, suggestions, and questions:

  1. The title is too lengthy and should be shortened for clarity and conciseness.
  2. Please revise the last sentence in the abstract to clarify the practical implications of the suggested findings.
  3. The statement "The early stages of life represent a critical period for the colonization..." needs elaboration. It's a common assertion, but what are its specific implications? Shouldn't we also consider the importance of intestinal microflora in later stages for proper maintenance?
  4. Why specifically choose Bacillus licheniformis (B. licheniformis) over other probiotics? What distinguishes it?
  5. The final sentence of the abstract and the introduction need refinement to better articulate the research purpose.
  6. Given the breadth of the topic, the introduction feels overly brief. Please expand it to provide more context.
  7. The labels in Figure 1 are too small; they should be enlarged for better readability.
  8. "These results indicated that the ability of anti-inflammatory and..." This statement is overly general; please clarify and provide specifics.
  9. The sentence regarding metschnikovii in Figure 3F is unclear and needs revision.
  10. Figures 3, 5, and 6 contain valuable information but are difficult to interpret due to small labels. Please enlarge the descriptions for clarity.
  11. "It is noteworthy that among the metabolites upregulated..." Please elaborate on the significance of these findings.
  12. The "Conclusions" section is too brief; please expand it to provide a more substantial summary of the study's outcomes.

Overall, the manuscript presents a compelling idea but requires substantial revisions. I look forward to reviewing it again after these improvements have been made.

Round 2

Reviewer 1 Report

Comments and Suggestions for Authors

The authors have revised the manuscript and answered all my comments. The article can be accepted for publication.

Reviewer 2 Report

Comments and Suggestions for Authors

Authors have significantly improved the article; it can be accepted.